# Oxylipins produced by *Pseudomonas aeruginosa* promote biofilm formation and virulence

Eriel Martínez[1] & Javier Campos-Gómez[1]

The oxygenation of unsaturated fatty acids by dioxygenases occurs in all kingdoms of life and produces physiologically important lipids called oxylipins. The biological roles of oxylipins have been extensively studied in animals, plants, algae and fungi, but remain largely unidentified in prokaryotes. The bacterium *Pseudomonas aeruginosa* displays a diol synthase activity that transforms several monounsaturated fatty acids into mono- and di-hydroxylated derivatives. Here we show that oxylipins derived from this activity inhibit flagellum-driven motility and upregulate type IV pilus-dependent twitching motility of *P. aeruginosa*. Consequently, these oxylipins promote bacterial organization in microcolonies, increasing the ability of *P. aeruginosa* to form biofilms *in vitro* and *in vivo* (in *Drosophila* flies). We also demonstrate that oxylipins produced by *P. aeruginosa* promote virulence in *Drosophila* flies and lettuce. Our study thus uncovers a role for prokaryotic oxylipins in the physiology and pathogenicity of bacteria.

[1] Southern Research, Department of Infectious Diseases, Drug Discovery Division, 2000 Ninth Ave South, Birmingham, Alabama 35205, USA. Correspondence and requests for materials should be addressed to J.C.-G. (email: jcampos-gomez@southernresearch.org).

The oxygenation of fatty acids is one of the main biochemical reactions in the synthesis of lipid mediators. Oxygenated fatty acids are the starting points for the synthesis of a great variety of biologically significant metabolites known as oxylipins. The best studied oxylipins are the leukotrienes and prostanoids in mammals, which are implicated in inflammation, fever, pain and other physiological processes[1]. The oxylipin jasmonic acid and its derivatives have been intensely studied in plants[2]. These compounds mediate hormone-like functions and are also involved in defence responses and development[3]. In algae and fungi, oxylipins participate in defence, reproduction and pathogenesis[4]. However, very little is known about the role of oxylipins in prokaryotes.

Oxylipin synthesis is mainly catalysed by fatty acid dioxygenases and monooxygenases, although they can also be produced by non-enzymatic chemical oxidation of fatty acids[5]. Fatty acid dioxygenases include lipoxygenases, cyclooxygenases, α-dioxygenases and diol-synthases, which produce fatty acid-derived hydroperoxides or endoperoxides[6]. For a long time no evidence existed for fatty acid dioxygenases in bacteria. However, in the last decade, in part due to the benefits provided by deep sequencing techniques, genes encoding putative fatty acid dioxygenases have been identified in the chromosome of many bacterial species. LoxA of *Pseudomonas aeruginosa* was the first prokaryotic dioxygenase to be characterized[7]. The enzyme transforms the arachidonic acid into 15-HETE. The role of LoxA remains understudied *in vivo*; however, given the known anti-inflammatory properties of 15-HETE it has been speculated it could play a role during infection by controlling the immune defences of the hosts[7].

*P. aeruginosa* also contains a fatty acid diol synthase activity that catalyses dioxygenation of several monounsaturated fatty acids, generating mono- and di-hydroxylated derivatives[8]. When oleic acid is used as substrate, the activity generates (10S)-hydroxy-(8E)-octadecenoic acid (10-HOME) and 7S,10S-dihydroxy-(8E)-octadecenoic acid (7,10-DiHOME) derivatives (Fig. 1). Given the biological relevance of oxylipins in other taxonomical groups, we hypothesized that oxylipins derived from bacterial metabolism may also have an important physiological meaning. To address this issue, we explored the role of the diol synthase-derived oxylipins in this important opportunistic pathogen. We found that oxylipins inhibit flagellum-driven swimming and swarming motilities but upregulate type IV pilus-dependent twitching motility of *P. aeruginosa*. Consequently, these compounds promote this bacterium's ability to form biofilms *in vitro* and *in vivo* (in the *Drosophila* fly model). We also demonstrate that oxylipins produced by *P. aeruginosa* promote its virulence in *Drosophila* flies and lettuce.

## Results

**Oxylipins regulate bacterial motility.** The diol synthase activity in the model strain *P. aeruginosa* PAO1 relies on a pair of genes, PA2077 and PA2078, that are members of di-heme-cytochrome C peroxidase[9,10]. To elucidate a possible role of these diol synthase-derived oxylipins, we introduced an in-frame deletion encompassing both PA2077 and PA2078 genes (Supplementary Fig. 1A). As expected, the resulting strain, which we called ΔDS, is a deficient 10-HOME and 7,10-DiHOME producer (Supplementary Fig. 1B). ΔDS displayed a normal growth rate in M63 complete medium supplemented with oleic acid up to 1 mg ml$^{-1}$, suggesting that this activity is not directly involved in the primary metabolism of the bacterium (Supplementary Fig. 2).

We then tested the effect of the diol synthase-derived oxylipins on the different types of bacterial motility. We first observed a negative effect of oleic acid on the swarming motility of PAO1.

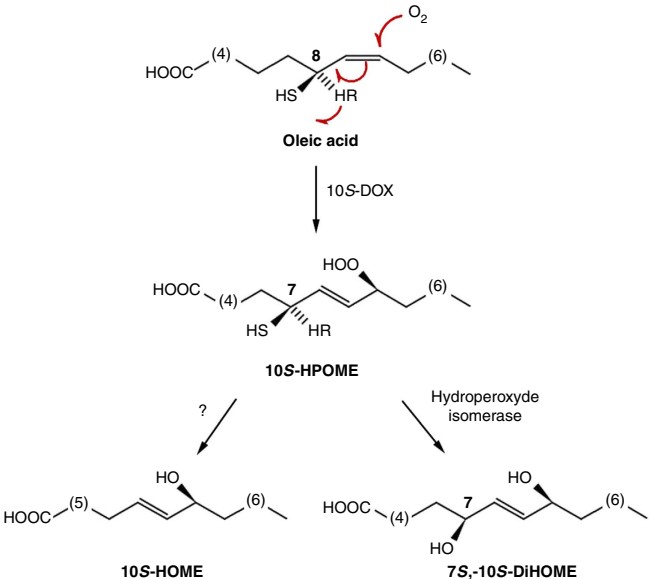

**Figure 1 | Biosynthetic pathway of oxylipins derived from the diol synthase activity of *P. aeruginosa* over oleic acid.** Oleic acid is first dioxygenated at position C10 by the enzyme 10(S)-dioxygenase (10S-DOX) (ref. 8). Subsequently, the hydroperoxide derivative (10S-HPOME) could be isomerized by the enzyme (7S,10S)-hydroperoxide isomerase to form 7S,10S-DiHOME or be reduced to 10-HOME by an undefined mechanism[8].

We found that such effect was largely dependent on the diol synthase activity, since no inhibitory effect of oleic acid on swarming was observed in ΔDS (Fig. 2a). To confirm oxylipins derived from the diol synthase activity on oleic acid were responsible for the negative effect on swarming, we purified 10-HOME and 7,10-DiHOME from culture supernatants of PAO1 grown in the presence of oleic acid (Supplementary Fig. 3). Increasing concentrations of both 10-HOME and 7,10-DiHOME significantly reduced ΔDS's ability to swarm, although the inhibitory effect of 10-HOME was notably stronger than that observed for 7,10-DiHOME (Fig. 2b).

Swarming is strongly dependent on a functional flagellum. Thus we investigated whether oxylipins were directly regulating flagellum motility by testing the effect of purified oxylipins on the swimming ability of ΔDS. Both 10-HOME and 7,10-DiHOME negatively affected ΔDS swimming motility. Consistent with the stronger effect of 10-HOME on swarming, the effect of 10-HOME on swimming was higher than that observed for 7,10-DiHOME (Fig. 2c).

In addition to flagellum-dependent motilities, *P. aeruginosa* displays a type IV pilus-driven twitching motility[11]. Surprisingly, we found that contrary to the negative effect of 10-HOME and 7,10-DiHOME in swarming and swimming, both oxylipins strongly promoted *P. aeruginosa* twitching motility (Fig. 2d). Increasing concentrations of 10-HOME or 7,10-DiHOME resulted in a direct proportional increase of ΔDS twitching motility. In addition, we directly observed the effect of oxylipins on twitching under the microscope. When supplemented with 10-HOME, ΔDS's expanding edges consistently projected longer and thicker branches, which expanded faster ($53.6 \pm 10.1$ pxl min$^{-1}$) than the controls without oxylipins ($22.0 \pm 11.5$ pxl min$^{-1}$) (Supplementary Movies 1 and 2).

**Oxylipins promote biofilm formation over abiotic surfaces.** Bacterial motility has a direct impact on the capacity of most bacteria to grow attached to surfaces as biofilms. During biofilm

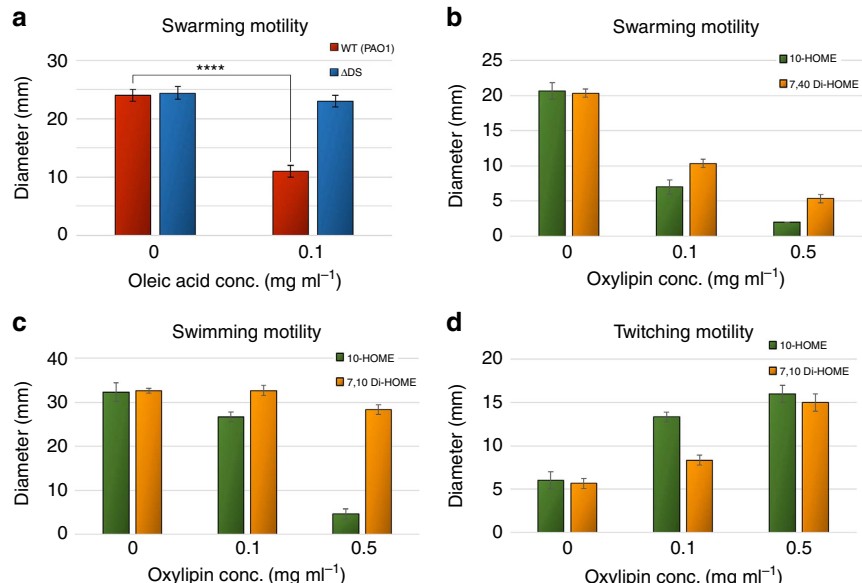

**Figure 2 | Oxylipins inversely regulate flagellum and type IV pilus associated motilities.** (**a**) Effect of adding oleic acid on the swarming motility of PAO1 and its ΔDS-derived mutant. Oleic acid at 0.1 mg ml$^{-1}$ inhibited PAO1 significantly (t-test, $P < 0.0001$), but not ΔDS swarming. (**b**) Both 10-HOME and 7,10 Di-HOME oxylipins inhibit swarming motility of ΔDS significantly (one-way ANOVA, $P < 0.0001$). (**c**) 10-HOME oxylipin strongly inhibits flagellum-driven motility of ΔDS strain (one-way ANOVA, $P < 0.0001$), but 7,10 Di-HOME has a weak effect ($P < 0.05$). (**d**) 10-HOME and 7,10 Di-HOME significantly increased ΔDS twitching motility (one-way ANOVA, $P < 0.0001$). Results from three independent experiments with $N = 3$ each. Error bars represent s.d.s.

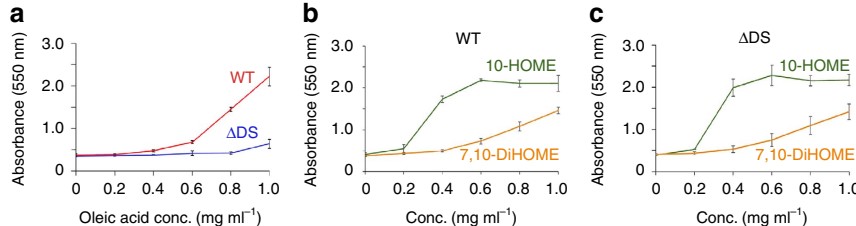

**Figure 3 | Oxylipins induce biofilm formation in *P. aeruginosa*** (**a**) Quantitation of the oleic acid effect on biofilm formation by the WT and ΔDS strains. Oleic acid induced a threefold increase in biofilm formation by WT compared to ΔDS. (**b**) Quantitative effect of 10-HOME and 7,10-DiHOME on biofilm formation by PAO1. 10-HOME showed a stronger effect than 7,10-DiHOME. (**c**) As expected, the effect of oxylipins on ΔDS biofilm formation was similar. Means and s.d. are from at least three independent experiments.

formation, *P. aeruginosa*, which has become a common model to study this process[12], transits from a free-swimming (planktonic) to a sessile phenotype[13]. In order to know if the effect of the diol synthase-derived oxylipins on *P. aeruginosa* motility has consequences in the ability of this bacterium to form biofilms, we compared the capacity of PAO1 versus ΔDS to form biofilms in polystyrene microtiter plates. We found that increasing concentrations of oleic acid up to 1 mg ml$^{-1}$ proportionally increased the capacity of PAO1 to form biofilm (Fig. 3a). Interestingly, such an effect was very weak on the ΔDS mutant, suggesting that oxylipins derived from the diol synthase activity were responsible for promoting *P. aeruginosa*'s biofilm formation *in vitro* (Fig. 3a). To confirm this, we tested the effect of pure 10-HOME and 7,10-DiHOME on the ability of *P. aeruginosa* to form biofilm on microtiter plates. Both 10-HOME and 7,10-DiHOME induced biofilm formation of PAO1 or ΔDS strains (Fig. 3b,c). Consistent with the stronger effect of 10-HOME on motility compared to 7,10-DiHOME, the positive effect of 10-HOME was also visibly higher than that of 7,10-DiHOME in inducing biofilm formation (Fig. 3b,c). For example, while 10-HOME achieved saturation at 0.6 mg ml$^{-1}$, the effect of 7,10-DiHOME just started to show at that same concentration (Fig. 3b,c).

Inverse regulation of flagellum- and type IV pilus-based motilities was previously described to promote formation of bacterial microcolonies[14,15], a first step in the organization of a biofilm. Thus, we explored whether oxylipins were inducing initiation of a biofilm *in vitro* by upregulating the formation of microcolonies. To this end, PAO1 bacteria constitutively expressing green fluorescent protein (GFP) were directly visualized by fluorescent microscopy. After 3 h of incubation in the absence of oxylipins, attached bacteria were distributed almost homogeneously over the surface (Fig. 4a, left panels). However, when the medium was supplemented with 10-HOME or 7,10-DiHOME at 0.4 mg mg$^{-1}$ and 0.8 mg ml$^{-1}$ respectively, both oxylipins consistently promoted bacterial organization in microcolonies (Fig. 4a, panels to the right). Again, the effect of 10-HOME in inducing microcolonies formation was visibly higher than that of 7,10-DiHOME. It started to be seen at 0.2 mg ml$^{-1}$ (Fig. 4a, centre panel) and quantity and size of microcolonies at 0.4 mg ml$^{-1}$ were higher even when it was used at half the concentration of 7,10-DiHOME.

We additionally explored whether microcolonies induced by oxylipins contained extracellular DNA (eDNA) and exopolysaccharide (EPS), which are characteristic components of the matrix

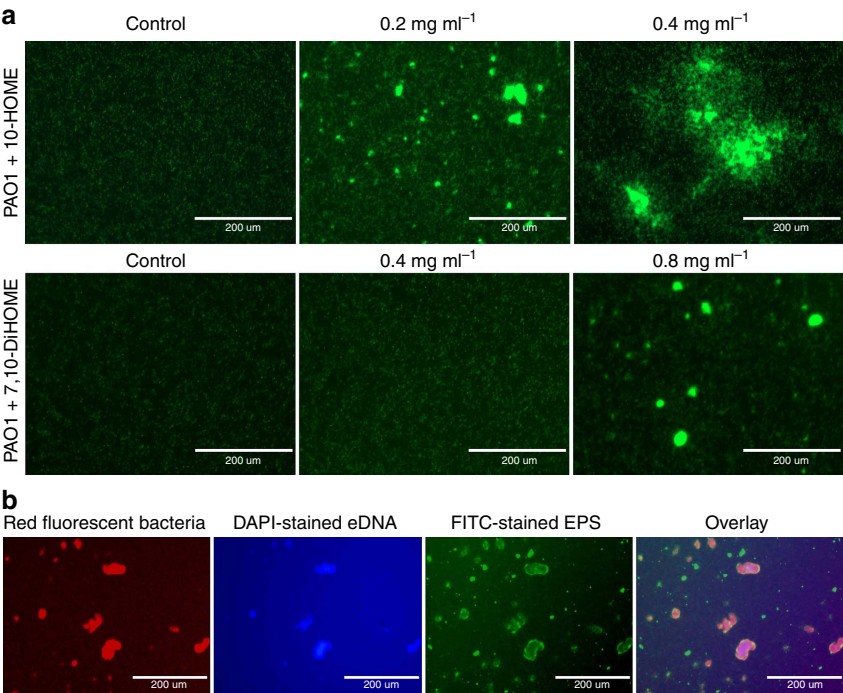

**Figure 4 | Oxylipins induce microcolony and biofilm formation over abiotic surfaces. (a)** Fluorescence microscopy pictures of GFP-expressing PAO1 on microtiter plates at different concentrations of 10-HOME (upper panels) or 7,10-DiHOME (lower panels). **(b)** Oxylipin-induced-microcolonies produce eDNA and EPS into the extracellular matrix. Microcolonies of RFP-expressing-PAO1, induced by 10-HOME at $0.2\,mg\,ml^{-1}$ (left panel), stained positive with 4',6-diamidino-2-phenylindole (blue) and concanavalin A-FITC conjugate (green), indicating the presence of both compounds in the matrix. In **a,b** bacteria were added at $10^5$ per well and pictures were taken 3 h after the addition of bacteria. Bars represent 200 μm. The size/resolution for each panel was adjusted from $7.770 \times 13.333$ in/72 dpi of the originals to $2.125 \times 1.550$ in/600 dpi in **a** and $1.760 \times 1.252$/600 dpi in **b**. Pictures are representative of three independent experiments with three replicates each.

of microcolonies deriving into biofilms[16]. DNA- and EPS-specific fluorescence staining using 4',6-diamidino-2-phenylindole and concanavalin A-FITC conjugate, respectively, showed the presence of abundant eDNA and EPS associated to the extracellular matrix of the microcolonies (Fig. 4b).

**Non-oxylipin-producing bacteria behave as social cheaters.** Bacteria usually produce extracellular factors required for bacterial growth and persistence. These factors benefit the whole bacterial population including cell variants that exploit them by 'cheating', avoiding the cost of production[17]. The diol synthase activity of *P. aeruginosa* occurs in the periplasm[18]. However, the resulting oxylipins cross the bacterial outer membrane and accumulate in the extracellular media. This prompted us to test whether the ΔDS mutant can profit from the extracellular oxylipins produced by the wild type (WT) by cheating to complement its biofilm deficiency when they are co-cultured. For this, PAO1 and ΔDS, constitutively expressing GFP or m-Cherry were co-cultured in minimal media M63 supplemented with oleic acid at $1\,mg\,ml^{-1}$. Quantitative analysis of bacteria incorporated into the biofilm revealed that, in coculture with PAO1, ΔDS efficiently incorporated into the biofilm. This suggested ΔDS cheated on oxylipins produced by the WT (Supplementary Fig. 4).

**Oxylipins induce biofilm formation over A549 cells *in vitro*.** We subsequently tested whether the effect of oxylipins on biofilm formation also had an impact in the ability of *P. aeruginosa* to colonize biotic surfaces. As *P. aeruginosa* infections are frequently associated with the respiratory tract[19], we tested the capacity of 10-HOME and 7,10-DiHOME to promote biofilm formation over

monolayers of A549 human alveolar epithelial cells. PAO1 expressing GFP was added to the cell monolayers (approximately 20 bacteria per cell) in the presence or absence of 10-HOME or 7,10-DiHOME. As observed in abiotic surfaces, both 10-HOME and 7,10-DiHOME oxylipins promoted microcolony formation of PAO1 over A549 monolayers. Once again, the effect of 10-HOME was consistently higher than that of 7,10-DiHOME when used at the same concentration (Fig. 5).

**Oxylipins promote biofilm formation *in vivo*.** We next investigated the effect of 10-HOME or 7,10-DiHOME in biofilm formation *in vivo*. The crop of *Drosophila melanogaster* orally inoculated with *P. aeruginosa* has been proposed as a model to visualize biofilm formation *in vivo*[20]. We orally inoculated flies with PAO1 or ΔDS expressing GFP in M63 complete medium. Twenty hours post infection the flies were killed and dissected crops were analysed by microscopy. Under these conditions no difference was observed between PAO1 and ΔDS. Both strains produced small microcolonies that were almost homogeneously distributed throughout the crops (Fig. 6a and Supplementary Fig. 5A). This result suggested that oxylipins were not produced under these conditions, probably by the lack of fatty acids availability in the crop compartment. However, when fly food was supplemented with 10-HOME or 7,10-DiHOME, both strains formed visibly larger microcolonies and biofilm covering greater area over the luminal epithelium of crops (Fig. 6b,c and Supplementary Fig. 5B,C). These observations suggest that *P. aeruginosa* can initiate biofilms *in vivo* over epithelial tissues, provided it has a source of fatty acids available for the synthesis of oxylipins.

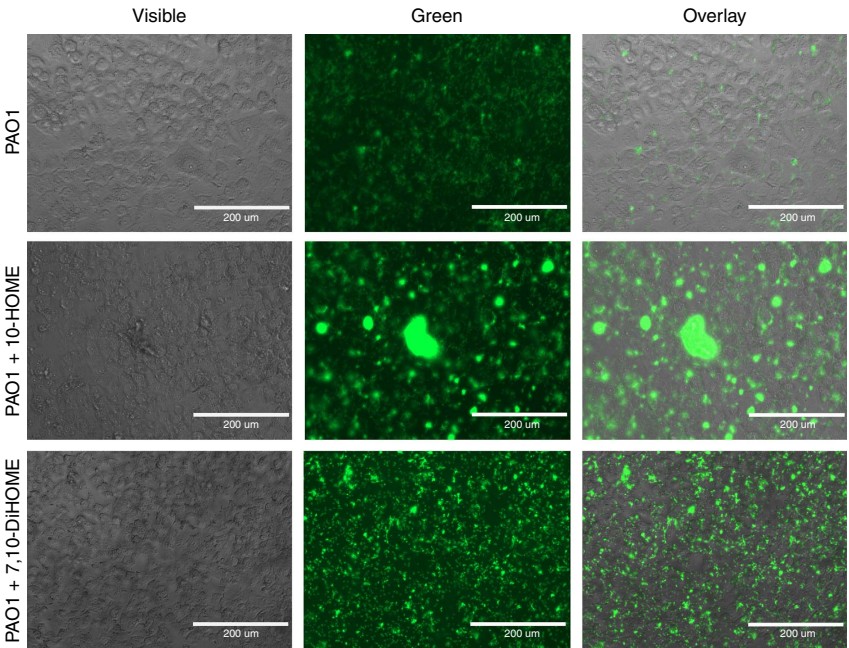

**Figure 5 | Oxylipins induce microcolony and biofilm formation over human epithelial cell surfaces.** GFP-expressing PAO1 was added to cell monolayers of A549 cells in the absence (upper panels) or presence of 10-HOME and 7,10-DiHOME (lower panels). Oxylipins were added at 0.4 mg ml$^{-1}$. Approximately 20 bacteria per human cell were added and pictures were taken 3 h after the addition of bacteria. Bars represent 200 μm. The size/resolution for each panel was adjusted to 2.125 × 1.550 in/600 dpi from 7.770 × 13.333 in/72 dpi of the originals. Pictures are representative of three independent experiments with three replicates each.

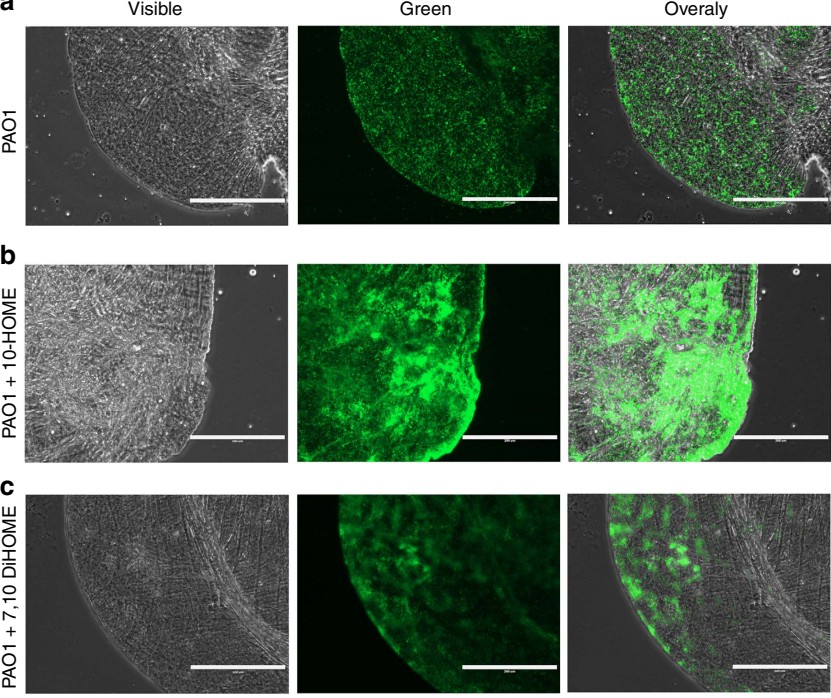

**Figure 6 | Oxylipins promote biofilm formation in _D. melanogaster_ crops.** Fluorescence microscopy pictures of dissected crops from _Drosophila_ flies fed with M63 media supplemented with 10-HOME or 7,10-DiHOME, into which GFP-expressing PAO1 strain was inoculated. (**a**) PAO1 formed few microcolonies and the bacteria were mostly homogeneously distributed all over the crop's lumen. However, when the media was supplemented with 10-HOME (**b**) or 7,10-DiHOME (**c**) PAO1 was able to abundantly form microcolonies and early biofilm in flies' crops. Bars represent 200 μm. The size/resolution for each panel was adjusted to 2.125 × 1.587 in/600 dpi from 17.770 × 13.333 in/72 dpi of the originals. Pictures are representative of two independent experiments, in which five flies were dissected in each case.

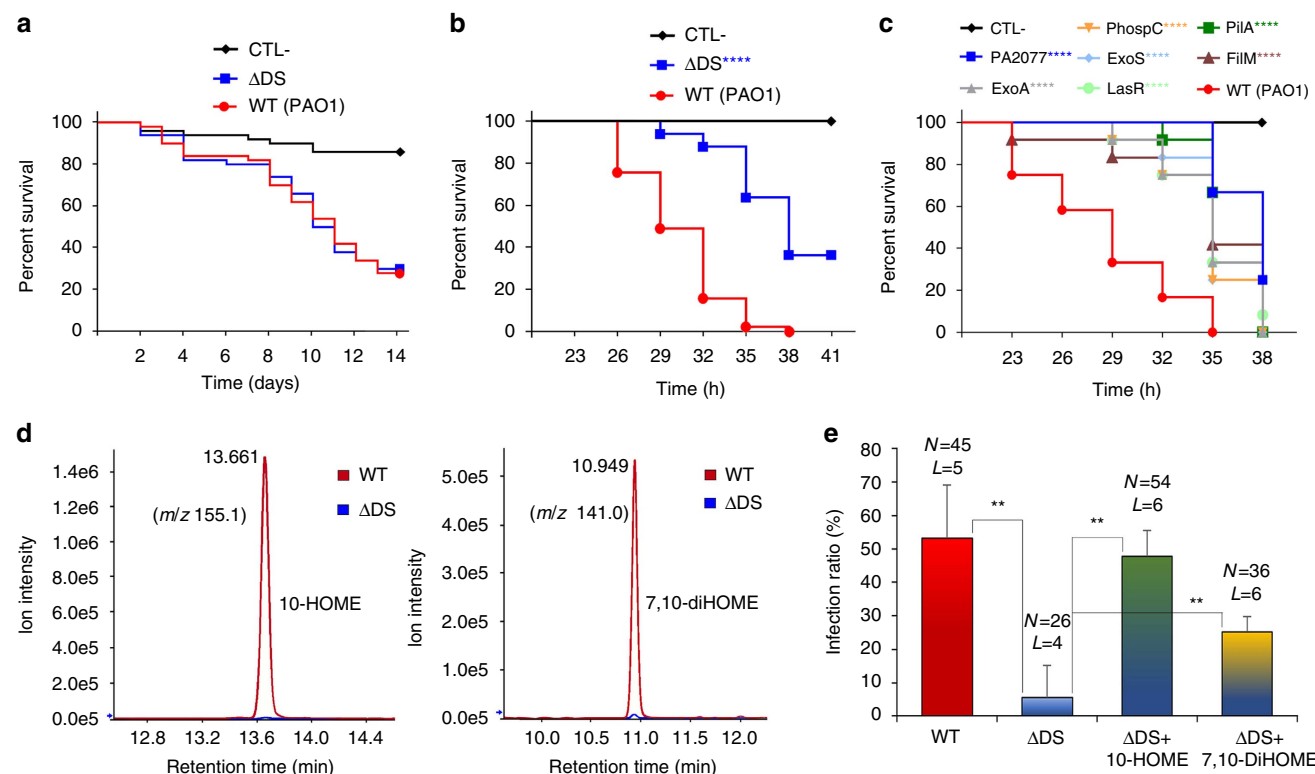

**Figure 7 | Oxylipins promote *P. aeruginosa* virulence in *drosophila* flies and lettuce. (a)** Kaplan-Meier plots of a survival experiment of *drosophila* flies orally inoculated with PAO1 or ΔDS did not show significant differences in survival between both strains (Mantel-Cox test; P = 0.9191). The graph corresponds to a single representative experiment of a total of four independent experiments done (each using 50 flies per condition). (**b**) Kaplan-Meier plots of a survival experiment of *drosophila* flies inoculated with PAO1 or ΔDS showed a significant attenuation of ΔDS (Mantel-Cox test; P < 0.0001, three independent experiments, each using 15 flies per condition: total 45 flies for each strain). (**c**) A survival experiment as the one shown in **b** but comparing seven transposon mutants including PA2077 and six well-established virulence factors of *P. aeruginosa*: ExoA, exotoxin; PhospC, phospholipase C; ExoS, exoenzyme S; LasR, quorum sensing transcriptional regulator; PilA, type 4 fimbrial precursor and FliM, flagellar motor switch protein. PA2077 and the rest of Tn mutants were not significantly different between them, but they all were significantly different from PAO1 (Mantel-Cox test, P < 0.0001, 15 flies per strain). (**d**) LC/MS/MS spectrometry analysis homogenates of flies infected with PAO1 or ΔDS. The reconstructed ion chromatograms show detection of 10-HOME (m/z 155.1, left graph) and 7,10-DiHOME (m/z 141, right graph) in flies infected with PAO1 (shown in red), but not in those inoculated with ΔDS (shown in blue). (**e**) Graph of an infection experiment done in lettuce. The percent of established infections over the total number of inoculation events is graphed. Means were significantly different (ordinary one-way ANOVA, **, P < 0.01, three independent experiments, error bars are s.d.). N, total of inoculation events; L, number of independent leaves inoculated.

**Oxylipins promote *P. aeruginosa* virulence in *Drosophila*.** Further, we assessed whether the effect of oxylipins on motility, and consequently on biofilm formation, correlated with the capacity of *P. aeruginosa* to colonize and develop an infection process *in vivo*. For this, we evaluated the virulence of PAO1 versus ΔDS against *D. melanogaster* through the oral and thoracic routes of inoculation, which are established models to study *P. aeruginosa* pathogenesis[20,21]. After several attempts we were unable to find any significant difference between the WT and the ΔDS mutant using the feeding model of inoculation (Fig. 7a). However, using the pricking method of bacterial inoculation we found that virulence of ΔDS was significantly attenuated in *Drosophila* flies. While PAO1 caused 100% of mortality 35 h post-inoculation, ΔDS was approximately half as virulent as PAO1 at the same time point (Fig. 7b).

In investigating the reasons for the different results using the feeding or pricking model of infection we hypothesized that this was probably because *P. aeruginosa* cannot acquire free fatty acids from the intact digestive tract of the flies as it does from the wounded tissues in the pricking model. To test this hypothesis we evaluated the presence of oxylipins in homogenates of flies infected by both routes of inoculation. We performed organic extractions of homogenates of flies inoculated with PAO1 or ΔDS

and analysed the extracts by thin layer chromatography (TLC). We were unable to detect 10-HOME or 7,10-DiHOME in homogenates of flies orally inoculated with PAO1. However, when the flies were inoculated by the pricking method we were able to detect the presence of two differential compounds in the samples of flies inoculated with PAO1 but not in those inoculated with ΔDS in the TLC analysis (Supplementary Fig. 6). Such compounds migrated with the same polarity of 10-HOME and 7,10-DiHOME, respectively (Supplementary Fig. 6). Subsequently, a liquid chromatography/mass spectrometry (HPLC/MS) analysis of the samples confirmed that 10-HOME and 7,10-DiHOME were present in the samples of flies inoculated by pricking with PAO1, but as expected not in those inoculated with ΔDS (Fig. 7d). This observation strongly suggests a causal relationship between oxylipins production by *P. aeruginosa* and its virulence in *D. melanogaster*.

To gain a relative measure of the extent of the attenuation caused by the disruption of the diol synthase activity when the virulence is tested by the pricking method, we evaluated in parallel the virulence of a mutant with a transposon inserted in PA2077 (impaired for oxylipins production) and other transposon mutants of genes encoding well-known virulence factors of *P. aeruginosa*. Interestingly, PA2077 showed similar level of

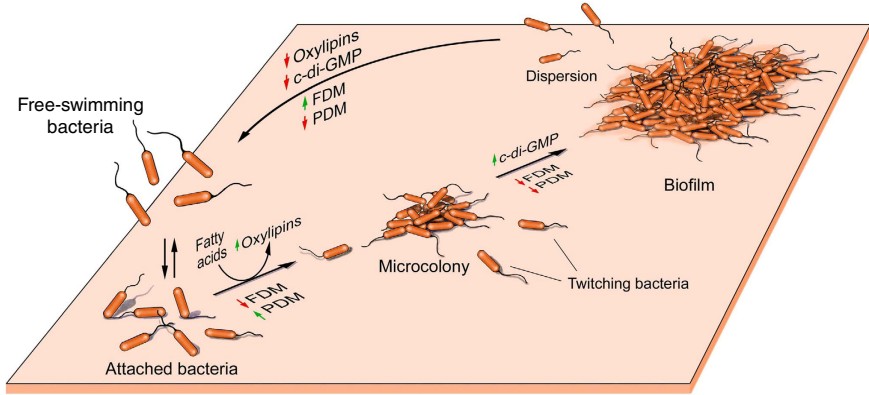

FDM, flagellum-driven motility; PDM, pilus IV-driven motility

**Figure 8 | Hypothetical model of the role of oxylipins in the formation of bacterial biofilms.** We propose that *Pseudomonas aeruginosa* senses fatty acids *in vivo* and transforms them into oxylipins, which induce twitching motility that in turn promotes microcolony formation. When sufficient number of bacteria is accumulated in the microcolonies, the quorum sensing mechanism is triggered, inducing signalling molecules as the c-di-GMP that, among other regulatory effects, promote biofilm formation by downregulating flagellum- and pilus IV-driven motilities.

attenuation to mutants of the most important virulence factors of *P. aeruginosa,* such as the exotoxin A, lysozyme S, phospholipase C, flagellum motility or type IV pilus (Fig. 7c).

**Oxylipins promote *P. aeruginosa* virulence in lettuce**. *P. aeruginosa* is also a common pathogen of plants. Thus, we compared the ability of PAO1 versus ΔDS to develop an infection process in lettuce leaves, a model of *P. aeruginosa* plant infection[22]. While PAO1 was able to establish an infection in more than 50% of the inoculation sites, ΔDS could establish infection only in 5% of the inoculation events. However, when ΔDS was inoculated with 10-HOME or 7,10-DiHOME the infection ratio increased to 48 and 25%, respectively (Fig. 7e). These results suggest that oxylipins also promote *P. aeruginosa* colonization of plant tissues.

## Discussion

Bacterial biofilms are widely recognized to play an important role in pathogenesis during bacterial infections[23]. Although the biology of biofilm has been extensively studied, little is understood about the signals governing the initiation of biofilms *in vivo*. Our results provide strong evidence suggesting that *P. aeruginosa* (and probably other bacteria as well) transforms fatty acids scavenged from the host into oxylipins as a way to sense the host environment and promote initiation of a biofilm lifestyle. We show that two oxylipins, 10-HOME and 7,10-DiHOME derived from a diol synthase activity of *P. aeruginosa* oleic acid are directly involved in this process. These oxylipins seem to play a critical role in the initial stages of biofilm formation by inducing a microcolony organization of attached bacteria. The mechanism by which 10-HOME and 7,10-DiHOME mediate this process includes the inverse regulation of flagellum- and type IV pilus-dependent motilities. The induced microcolonies can subsequently lead to a mature biofilm, a process involving induction of the second messenger c-di-GMP that, among other functions, inhibit both flagellum- and type IV pilus-dependent motilities[24] (Fig. 8).

Although our results suggest both 10-HOME and 7,10-DiHOME oxylipins act on a common physiological pathway, the effect of 10-HOME on biofilm formation was consistently higher than that of 7,10-DiHOME. We currently do not know why *P. aeruginosa* produces two metabolites with a redundant function; however, the kinetic study of appearance/disappearance of 10-HOME and 7,10-DiHOME in *P. aeruginosa* culture supernatants showed that while 10-HOME appears first in time, 7,10-DiHOME remains longer in the stationary phase of the culture (Supplementary Fig. 7). Based on this observation it is tempting to speculate that 10-HOME could be responsible for producing an abrupt transient switch from a planktonic to a sessile behaviour at the very beginning phase of the biofilm formation once fatty acids are sensed. On the other hand, 7,10-DiHOME, with a more moderate activity, may prolong such effect during later states of biofilm maturation.

We want to remark here that we found an effect of oxylipins on virulence in *Drosophila* inoculated by pricking the thoracic segment of the flies, which implies tissue damage, but not in flies inoculated by the oral route. This is consistent with our observation that PAO1 and ΔDS similarly colonized the fly crops in the absence of oxylipins using the oral route (Fig. 6a and Supplementary Fig. 5A). Outstandingly, exogenous oleic acid in the inocula was not required to observe an attenuated virulence of ΔDS compared to PAO1 in *D. melanogaster* by the pricking method. This suggests that *P. aeruginosa* inoculated by the pricking method have access to a source of host fatty acids in wounded tissues but not in the intact digestive tract of flies inoculated by the oral route (at least under the conditions tested). These fatty acids could serve as substrates for oxylipin synthesis, which this bacterium uses for its own benefit.

An inevitable arising question is whether the diol synthase is active during infections of organisms closer to humans, like other mammals. A previous *Drosophila*-based screening to determine virulence factors of *P. aeruginosa* PA14 revealed that a transposon insertion into the PA2077 gene not only attenuated PA14 virulence in the fly, but also in a murine model of peritonitis[25]. Another study focused on gene expression and fitness determinants during acute and chronic infections of murine models with *P. aeruginosa*, revealed that genes encoding the diol synthase activity, PA2077 and PA2078, are overexpressed in both acute and chronic infections (18.90 and 16.23 times in acute versus 36.20 and 29.47 times in chronic, respectively) compared to a non-infective well-defined *in vitro* condition[26]. Additionally, they found that PA2077 and PA2078 genes were important for bacterial fitness[26]. These studies, which are in agreement with our results, represent an independent corroboration of the impact caused by disrupting the diol synthase activity on *P. aeruginosa* pathogenicity and additionally reveal a possible role of oxylipins production during *P. aeruginosa* colonization of mammalian tissues.

The wide distribution of oleic acid among human tissues also supports the feasibility of oxylipins production during *P. aeruginosa* infections. Oleic acid is the most abundant fatty acid in human adipose tissue[27] and second in abundance in human tissues in general[28] including the skin, where it functions as an important component of the innate immune response[29]. Particularly, in cystic fibrosis patients, the alveoli secretion is enriched in host-derived lipids triggered by the host immune response[30], which might be a source of fatty acids for the production of oxylipins. Additionally, these patients exhibit an increased bronchial cytosolic phospholipase (cPLA) A2α activity, which hydrolyses membrane phospholipids at the sn-2 position, releasing free fatty acids[31]. Interestingly, specific inhibition of cPLA A2α reduces mouse mortality induced by *P. aeruginosa* pulmonary infection[32], which could, at least in part, be due to the reduced availability of fatty acids for oxylipin production.

While our experiments suggest that the motility phenotype of ΔDS and its impaired ability to form biofilm directly affect its virulence in *Drosophila* and lettuce, an additional role of oxylipins in the context of bacterial-host interaction should not be discarded. In animals, plants and fungi, oxylipins are signalling molecules involved in cross kingdom communication[33]. Thus, a role of prokaryotic oxylipins in mediating bacterial-host cross communication should be seriously considered. Altogether, the results presented here provide strong evidence supporting that prokaryote oxylipins, as those produced by eukaryotes, are important lipid mediators involved in critical steps of the bacterial biological cycle, such as regulation of biofilm lifestyle and virulence.

## Methods

**Strains and culture conditions.** *P. aeruginosa* strain PAO1 (obtained from the Manoil Lab at the University of Washington, Seattle, WA, USA) and its isogenic mutant ΔDS (diol synthase deletion mutant) were used throughout the entire study. Deletion of the diol synthase operon was performed by allelic exchange using the suicide vector pEX100Tlink and a previously described method[34] Briefly, a fragment of *P. aeruginosa* chromosome containing the genes PA2078 and PA2077 plus ∼300 bp of chromosomal flanking regions were amplified by PCR using the primers JCG1 (5′-ggcgaaagcttcgccttcctgccg-3′) and JCG2 (5′-ggcgg gaattctggtcaccaccttct-3′). These primers introduced sites EcoRI and HindIII at the ends of the amplified fragment that were used for cloning into the suicide vector pEX100Tlink to obtain the plasmid pEX-PA77-78. From this plasmid a SmaI-StuI fragment internal to the diol synthase operon was deleted to obtain the final construction pEXΔDS. This plasmid was used to make an in-frame deletion of PA2077-78, through allelic exchange in the chromosome of PAO1 (Supplementary Fig. 1A). The diol-synthase-activity-lacking strain obtained was named ΔDS. The mutant genotype was confirmed by PCR and sequencing, and its oxylipin-negative phenotype was confirmed by TLC (see below). Complementation of the mutant was performed by replacing the mutated allele with the original copy from the parental strain PAO1, also through allelic exchange. Green and red fluorescent *P. aeruginosa* were obtained by transformation with the plasmids pMF230 or pMF440, which constitutively express the GFP mut2 and mCherry fluorescent proteins, respectively. Plasmids pMF230 and pMF440 (Addgene plasmids # 62546 and 62550) were a gift from Michael Franklin (Montana State University). Other *P. aeruginosa* strains used were the transposon mutants of genes encoding exotoxin A (*exoA*, strain PW3079), phospholipase C (*plcN*, PW6586), exoenzyme S (*exoS*, PW7479), quorum sensing transcriptional regulator (*lasR*, PW3597), type 4 fimbrial precursor (*pilA*, PW8622) and the flagellar motor switch protein (*fliM*, PW3621), which were acquired from the transposon library collection of University of Washington. *Escherichia coli* DH5α (Invitrogen) was the host for plasmid constructions and *E. coli* S17-1 λpir (a gift from Jorge Benitez— Morehouse School of Medicine) was used as a donor strain for bacterial conjugation when required.

The strains were routinely grown in lysogeny broth (LB) medium at 30 °C, to which agar was added when solid medium was required. LB agar without NaCl plus 15% sucrose was used to segregate suicide plasmid from merodiploids during construction of ΔDS by allelic exchange. Biofilm formation was performed in M63 media supplemented with 2% glucose, 5% casaminoacid and MgSO4 1 mM (M63 complete). Antibiotics were added, when necessary, at the following concentrations: Ampicillin (Amp), 100 µg ml$^{-1}$; Carbenicillin (Cb), 300 µg ml$^{-1}$. Oleic acid 90% (Sigma 364525) was added to cultures for oxylipin production and purification. M63 complete media was supplemented with oleic acid 99% (Sigma O1008) or purified oxylipins when required for the study of biofilm formation over polystyrene surface of microtiter plates or over biotic surfaces.

**Thin layer chromatography.** TLCs were carried out on silica gel (60 Å) plates of 20 × 10 cm, 200 µm thickness (Whatman). A mix of hexane, ether and acetic acid (80/20/5) was used as mobile phase. TLC plates were revealed with 10% phos- phomolybdic acid in ethanol.

**Purification of 10-HOME and 7,10-DiHOME oxylipins.** The supernatant of a 500 ml PAO1 culture grown in M63 complete supplemented with 1% oleic acid was used to purify oxylipins derived from the diol synthase activity. The culture was centrifuged at 8000*g* for 15 min, then the supernatant was recovered and acidified (pH = 2) with hydrochloric acid. Then a one vol/vol organic extraction was performed using ethyl acetate. The organic phase was evaporated and the dried mixture obtained was dissolved in 3 ml of ethyl acetate and used for purification of the oxylipins on an Isco Teledyne Combiflash Rf 200 with four channels with 340CF ELSD (evaporative light scattering detector). Universal RediSep solid sample loading pre-packed cartridges (5.0 g silica) were used to absorb crude product and purified on 24 g silica RediSep Rf Gold Silica (20–40 µm spherical silica) columns using an increasing gradient of ethyl acetate (solvent B) over hexane (solvent A) (Supplementary Fig. 3A). Fractions collected for each detected peak were combined and evaporated, then dissolved in methanol. The purity of the oxylipins was checked by HPLC/MS analysis.

**HPLC/MS analysis.** Purified 7,10 Di-HOME and 10-HOME were prepared at 1 mg ml$^{-1}$ in methanol (stock solution), from which samples to be analysed were prepared by diluting in ddH2O with 0.1% formic acid. For each sample, a 20 µl injection was loaded onto a Synergi Hydro-RP 80A 250 × 2 mm C18 column (Phenomenex) using a Shimadzu Prominence System Binary Pump (Shimadzu Scientific Instruments, Inc., Columbia, MD, USA) at a flow rate of 350 µl min$^{-1}$ using ddH2O with 0.1% formic acid and acetonitrile with 0.1% formic acid for mobile phase A/B respectively. The gradient proceeded from 10 to 80%B over 11 min, then to 100%B at 14 min, then re-equilibrated back at initial conditions for 6 min for a total of 20 min per evaluation using the SCIEX 4000 Triple Quadrupole Mass Spectrometer (Concord, Ontario, Canada) in the ESI negative ion mode. Nitrogen was used as a nebulizer and curtain gas (CUR = 20). The collision gas, collision energy and temperature were set at 10 °C ( − 30 eV for 10-HOME, − 34 eV for 7,10-DiHOME) and 600 °C, respectively. GS1 and GS2 gases were set at 40 °C and 60 °C respectively. Analyst 1.6.2 software controlled the LC-MS/MS system.

**Motility assays.** Swimming, swarming and twitching motilities were studied using the methods described by Rashid and Kornberg[35]. To capture videos of twitching motility, 100 µl aliquots of the twitching medium[35] were deposited over microscope cover slips. Once the medium solidified, it was punctured in the centre up to the bottom with an extended inoculation loop embedded with a bacterial suspension (OD600 = 1) of the strain to be filmed. The cover slips were incubated for 6 h at 37 °C, then placed inverted on the microscope stage (Olympus BX53 system microscope) and the twitching motility filmed using an XM10 incorporated camera (Olympus) controlled by the software cellSens Standard 1.6 (Olympus).

**Quantification of biofilm formation.** Biofilm assays were performed following the O'Toole protocol[36]. Briefly, *P. aeruginosa* strains were cultured overnight in LB agar plates at 37 °C. Bacterial suspensions were prepared in M63 medium to an OD600 = 1. Ten microliters of bacterial suspension was inoculated into each well of a 96-well microtiter plate containing 200 µl of M63 complete media. Oleic acid or pure oxylipins were added to the medium at desired concentrations when required. Biofilms were allowed to form at 30 °C overnight. For biofilm quantification the wells were washed two times with 1 × PBS and 200 µl of 0.1% crystal violet was added to the wells and incubated for 10 min. Wells were washed three times with 1 × PBS, then crystal violet-stained biofilm was solubilized with 250 µl of 30% acetic acid and the absorbance was measured at 550 nm.

**Bacterial attachment to microtiter plates.** To study bacterial attachment and microcolony formation similar conditions to the previous section experiment were used, but using 10$^5$ bacteria per well constitutively expressing GFP from the plasmid pMF230. Subsequently, the plates were incubated for 3 h at 37 °C, observed under the fluorescence microscope and pictures were taken.

**Detection of eDNA and EPS in the extracellular matrix.** Microcolonies induced by the addition of 10-HOME or 7,10-DiHOME, obtained as described in the previous section, were stained with 4′,6-diamidino-2-phenylindole and con- canavalin A-FITC conjugate at 0.5 µg ml$^{-1}$ and 50 µg ml$^{-1}$, respectively. In this case *P. aeruginosa* constitutively expressing mCherry RFP from the plasmid pMF440, was used to distinguish green fluorescence of the FITC conjugate. After 30 min of incubation with the fluorescent compounds the wells were washed with PBS 1 × and observed under the fluorescence microscope.

**Biofilm assay over A549 monolayers.** A549 human alveolar epithelial cells (ATCC CCL-185) were plated into black clear bottom 96-well-tissue culture plates

at 20,000 cells per well in Dulbecco's modified Eagle medium supplemented with bovine fetal serum and glutamine and allowed to grow up to confluence. Cells were washed with PBS $1 \times$ and $100 \mu l$ of M63 complete medium—with or without oxylipins as required—were added to each well. GFP-expressing *P. aeruginosa* was added at 20 bacteria per A549 cell and incubated for 3 h. Plates were washed two times with PBS $1 \times$ and the cell monolayers with attached green fluorescent bacteria were observed and pictures captured with a fluorescence microscope (EVOS FL Cell Imaging System, Life Technologies) using the same settings for each picture. For presentation of figures all relevant pictures were assembled as panels into a single file using Adobe Photoshop. All comparable pictures were subjected to the same transformations during the process (resizing, resolution change), which is specified in each pertaining figure legend. This also applies to pictures of the next section (crop imaging).

**P. aeruginosa imaging inside Drosophila crops.** All experiments involving *D. melanogaster* were done using 3-day-old flies from both sexes of the WT Oregon R (acquired from Carolina Company). *P. aeruginosa* colonization of *D. melanogaster* crop was performed as previously described by Mulcahy et al.[20] *P. aeruginosa* strains constitutively expressing GFP were cultured on LB agar plates. Bacteria were resuspended in LB to OD600 = 1. Then $100 \mu l$ of the suspension was spotted onto a sterile filter (Whatman) that was placed on the surface of 5 ml of LB agar containing 5% sucrose. The medium was supplemented with 10-HOME or 7,10-DiHOME when required. Flies were allowed to grow under this condition for 20 h and then killed. Crops were placed on a drop of PBS on a microscope slide, sealed with a coverslip and observed using an EVOS FL Cell Imaging System. Pictures were captured using the same settings for each picture.

**Virulence assay in orally inoculated Drosophila.** *Drosophila* flies were inoculated with the tested strains as described in the previous section, but instead of killing the flies they were incubated at room temperature ($\sim 25 \,°C$) and fly survival was followed and recorded for 14 days.

**Virulence assay in Drosophila inoculated by pricking.** Flies were anaesthetized with FlyNap (Carolina) and pricked in the thorax using a needle (BD Ultra-Fine Nano Pen Needles 32 g 5/32 inch) that was loaded with the tested strain of *P. aeruginosa* by dipping in a bacterial suspension at $OD_{600} = 1.0$ in M63 medium. After inoculation, the flies were placed into a vial (15 flies per vial) containing Formula 4-24 Instant *Drosophila* Medium (Carolina). Flies were maintained at room temperature and survival was monitored and recorded up to 36 h post inoculation.

**Detection of oxylipins in P. aeruginosa infected flies.** Groups of 20 infected flies were homogenized using an Omni THQ homogenizer with disposable Omni Tips plastic generator probes (OMNI international) in 2 ml of PBS $1 \times$. The homogenates were centrifuged to eliminate fly and bacterial debris and total fatty acids were extracted as described above (see section Purification of diol synthase-derived oxylipins). Extracted samples were analysed first by TLC and then by HPLC/MS (see above sections for TLC and HPLC/MS analyses) to identify the presence of 10-HOME and 7,10-DiHOME. To determine the presence of the oxylipins, samples were analysed by the MRM method using mass transitions $m/z$ 297.3/155.1 for 10-HOME and 313.3/141.1 for 7,10-DiHOME.

**Lettuce infection.** Individual lettuce leaves taken from the external foliage of romaine hearts (*Lactuca sativa* L. var. *longifolia*) were inoculated at 2–3 cm apart intervals into the central nervure of each leaf with the *P. aeruginosa* strain to be tested (3 μl of a bacterial suspension at an $OD_{600} = 3$) or the inoculation vehicle alone (M63 medium) using a 25 5/8 G needle connected to a pipette. When required 10-HOME or 7,10-DiHOME, or both were added to the inocula or the control vehicle at a concentration of $0.5 \,mg \,ml^{-1}$). Inoculated leaves were placed in plastic beakers with the inferior part of the central nervure submerged in a solution of 10 mM $MgSO_4$ in water. The leaves were incubated for 5 days at room temperature. In the case of the leaves coinoculated with bacteria and oxylipins, 3 μl of each oxylipins at $0.5 \,mg \,ml^{-1}$ were deposited over the inoculation sites once a day up to day third. The vehicle alone or with oxylipins at the same concentrations used for bacterial inoculation were used as negative controls. At the end of the experiment the leaves were evaluated for necrosis around the inoculation points. A dark brownish necrotized area of more than 5 mm of diameter was recorded as a successful infective event.

**Statistical analysis.** Kaplan-Meier plots of *Drosophila* flies survival experiments were compared using the log-rank (Mantel-Cox) test. We used 15 flies per condition to be able to detect an effect size proportion of surviving subjects = 0.5 with 99% power using a type I error of 0.01 (calculated using GraphPad StatMate 2). For randomization of *Drosophila* the flies were anesthetized and distributed to the final experimental container disregarding sex or size. In the case of the lettuce leaves, we were careful to take only external leaves of romaine hearts (up to 2–3 layer of leaves were taken per lettuce plant). Leaves coming from

different plants were mixed and randomized, then the leaves were taken and distributed to the final containers. One lab member distributed and inoculated the flies or lettuce leaves and assigned letter codes to each container. Then other lab member counted dead flies and lettuce lesions. At the end of the experiments letter codes were reconciled with the respective experimental condition used. In the rest of the experiments one-way ANOVA or *t*-test (two-sided) were used as needed to calculate significant differences between means of different conditions after appreciating the variance was similar between groups being compared. All statistical analyses were performed using GraphPad Prism 7 software.

**Data availability.** The authors declare that the data supporting the findings of this study are available within the article and its supplementary information files, or from the corresponding author on request.

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

## Acknowledgements

We thank Maaike Everts (University of Alabama at Birmingham—UAB) and Timothy Sellati (Southern Research—SR) for critical review of the manuscript. We also thank Vibha Pathak, Phanindra Venukadasula and Vandana V. Gupta (SR) for their assistance in purifying the oxylipins and Landon S. Wilson and Taylor F. Berryhill (UAB) for LC/MS analysis of oxylipins. We also thank Michael Franklin for providing plasmids pMF230 and pMF440 through Addgene and the Manoil Lab (University of Washington) for supplying the mutants from their defined transposon library collection, whose construction was funded by NIH grant P30 DK089507. This work was supported by intramural funds from Southern Research, by the Alabama Drug Discovery Alliance (a collaboration between the University of Alabama at Birmingham and Southern Research) and by the National Center for Advancing Translational Sciences of the National Institutes of Health under award number UL1TR001417 assigned to J.C.-G.

## Author contributions

E.M. and J.C.-G. designed, performed experiments and wrote the manuscript.

## Additional information

**Competing financial interests:** The authors declare no competing financial interests.

**How to cite this article**: Martínez, E. & Campos-Gómez, J. Oxylipins produced by *Pseudomonas aeruginosa* promote biofilm formation and virulence. *Nat. Commun.* **7**, 13823 doi: 10.1038/ncomms13823 (2016).

**Publisher's note**: 

