## [Peer Review File · Nature Communications]

Reviewers' comments:

Reviewer #1 (Remarks to the Author):

This study of the role of oxylipins in biofilm formation and pathogenesis in *Pseudomonas aeruginosa* is an exciting report of a novel signaling molecule that appears to play a very important role. These findings are very novel and highly significant to our understanding of how biofilms are formed during an infection, and therefore to the role of biofilms in infection. The paper was extremely well written, and was one of the most interesting papers I have read in the area of *Pseudomonas aeruginosa* and biofilms in a long time.

This group previously characterized the biochemical reactions of oxygenated fatty acids by *Pseudomonas aeruginosa* that was fed oleic acid. In this study, they identify and demonstrated the function of two enzymes that transform monounsaturated fatty acids into two species of oxylipins, which are synthesized in the periplasm and that diffuse back out of the cell. The mutants do not have growth defects in the presence of the substrate, indicating these enzymes are not required for primary metabolism. The oxylipins 10-HOME and 7,10-diHOME were purified and shown to reduce swarming/flagellar-dependent motility but to increase twitching/type IV pili-dependent motility. Since twitching is a surface motility that is known to influence biofilm formation, they demonstrated a role for oxylipins in biofilm formation on polystyrene surfaces. The double Δ DS mutant had restored biofilm formation on plastic surfaces, and on the surface of A549 monolayers, when supplied with exogenous 10-HOME and 7,10-diHOME. In all addback experiments, 10-HOME induced a more robust biofilm response.

The authors used a third in vivo biofilm model using the crop/GI biofilms formed when fruit flies are fed *Pseudomonas aeruginosa*. The in vivo biofilm results were identical to the in vitro biofilms, where addback of oxylipins restored and induced biofilm formation for the Δ DS mutant. Next, the authors showed that the oxylipin synthesis mutant was defective for virulence in the acute/pricking model of fruit fly infection. The virulence defect was comparable to other significant virulence factors.

It appears that monounsaturated fats were used in vivo during the infection, and converted to oxylipins during wt infections. These virulence phenotypes match those reported in previous publications also showing a role of these genes in acute and chronic mouse infections, as well as acute *Drosophila* infections.

The overall finding that *P. aeruginosa* scavenges fatty acids, a known and abundant food source in many infectious, and transforms fatty acids into a central signaling molecule for biofilm formation, provides a major leap in our understanding of how biofilms are formed in vivo.

Suggested improvements.

Why was virulence of the Δ DS mutant only tested in the acute fly infection model, when the biofilms were visualized in the feeding model? The feeding model should also be used to test the virulence of the double DS mutant. Although this kind of mutant may have increased virulence after feeding, this would be consistent with previous findings that mutants lacking EPS don't form biofilms in the crop and they disseminate and kill the fly faster. Although they may be "more virulent", this result would be consistent with the published model, and would further support this study.

For the above suggested experiment using the feeding fly model, the addition of oxylipins would restore the parental levels of killing, in addition to the restoration of biofilms in the crop.

The authors used one measure of biofilm formation to assess biofilms throughout, which is the presence of aggregates. It would strengthen the study to demonstrate that these aggregates had

other features of biofilms, such as the presence of EPS and eDNA in the matrix, or increased antibiotic tolerance, or they somehow contributed to immune evasion.

Since these reactions involving fatty acids require oxygen, would this pathway be operating in chronic lung infections in CF patients? CF lung infections are considered one of the best examples where biofilms contribute to a chronic infection.

Minor comments.

Incorrect spelling of "established", line 133

Reviewer #2 (Remarks to the Author):

In 2014 a Spanish group published a manuscript on the production and secretion of two oxylipins in the opportunistic pathogen *Pseudomonas aeruginosa* and they identified two genes that were responsible for their production. This manuscript is a follow-up study and aims at unravelling the functional role of oxylipin production in bacterial pathogenicity. The authors describe an impact of the oxylipins on motility and biofilm (in vitro and in vivo) phenotypes and provide convincing evidence that oxylipins are produced in vivo and contribute to bacterial virulence. They argue that the bacteria transform fatty acids scavenged from the host into oxygenated derivatives as a way to sense the host environment in order to induce appropriate bacterial responses. The finding that the two genes involved in oxylipin biosynthesis (PA2077 and 2078) are transcribed also under normal lab conditions (as can be seen in published transcriptional analysis) raises the question of whether *P. aeruginosa* is dependent on the external supply of fatty acids to produce oxylipins. Testing this would be important to validate the hypothesis.

This is a good manuscript that provides insight into the role of oxylipins in *P. aeruginosa* pathogenicity. However, there is no attempt to mechanistically unravel the link between oxylipin production and pathogenicity, and thus lacks broad appeal in its current state.

Reviewer #3 (Remarks to the Author):

In this manuscript the authors provide evidence for the role of two oxylipins, produced from oleic acid, in the ability of *P. aeruginosa* to produce microcolonies in vitro and in vivo and the subsequent effect on pathogenicity.

The role of microbial oxylipins in virulence is an understudied area and almost nothing is known of the role of bacterial oxylipins in the biology of microbes and especially in the infection process. As such these findings are highly novel and of interest to researchers in this particular field and to those studying virulence factors in general.

I have four suggestions to increase the strength of the conclusions:

1. The authors mention that they determined the concentration of oxylipins in the produced during infection of *Drosophila*, however they make no mention of this physiological concentration and it is unclear if this concentration has any significance when studying the effect in vitro. Are the in vitro concentrations used in the same range as the in vivo concentrations and if not, why not?
2. The authors should mention in their materials and methods section how many times an experiment was repeated and how the significance values were determined.
3. For the formation of microcolonies in vitro, the authors should include the wild type strain supplemented with oleic acid to compare the effect with the mutant. Here the physiological concentration of oxylipins may also be important when supplementing the mutant. Similarly, for the in vivo experiments, the wild type can be included in order to compare the extent of microcolony formation.

4. No results for bacterial attachment and kinetics (0 to 3 h) are presented.

Minor issue: For the oral inoculation of *Drosophila* – only the GFP-expressing strain's result is given, so the authors may consider omitting the mCherry from the materials and methods (Ln 316)

The authors should also look at some minor language issues:

Ln 71: Bacterial motility.....

Ln 75: "...motility has consequences..."

Ln 117: "...promoted microcolony formation..."

Ln 201: "...transform fatty acids scavenged..."

Ln 223: "For this plasmid..."

Ln 228-229: "...by replacing the mutated allele with the original..."

Ln 238: *Escherichia coli* has to be in italics

Ln 288: "...software Cell Sense..."

Ln 311: I do not think "suffered" is the correct word in this context

Ln 338: "...determine the concentrations of the analytes..."

Ln 357: *Drosophila* must be in italics

Reference 16 – *Pseudomonas* should be with a capital letter.

Signed: Carolina Pohl.

Point-by-point discussion of issues raised by the reviewers:

Answers to reviewer 1:

- R1:** Why was virulence of the delta DS mutant only tested in the acute fly infection model, when the biofilms were visualized in the feeding model? The feeding model should also be used to test the virulence of the double DS mutant. Although this kind of mutant may have increased virulence after feeding, this would be consistent with previous findings that mutants lacking EPS don't form biofilms in the crop and they disseminate and kill the fly faster. Although they may be "more virulent", this result would be consistent with the published model, and would further support this study.
- A:** This is a very logical concern of the reviewer. As a matter of fact we also evaluated the virulence in the feeding model, but after several attempts we didn't find any significant difference between the WT and the Δ DS mutant. In investigating the reasons for this result using the feeding model we noticed there was no difference between the WT and Δ DS mutant in terms of biofilm formation in fly crops (this result is now included in the revised manuscript). We then hypothesized that this was probably because *P. aeruginosa* cannot acquire free fatty acids from the intact digestive tract of the flies as it does from the wounded tissues in the pricking model. To test this hypothesis we evaluated the presence of oxylipins in homogenates of flies infected by the oral route. We were unable to detect 10-HOME or 7,10-DiHOME in flies orally inoculated with PAO1; unlike our ability to detect their presence in flies inoculated by pricking. Since there is no difference in the capacity of PAO1 and Δ DS to form a biofilm in the fly crops, neither oxylipins are produced in the digestive tract, we drew the conclusion that the feeding model is not useful to evaluate differences in virulence between PAO1 and Δ DS. As we are interested in precisely knowing the role of oxylipins in virulence we decided to exclude this result from the paper and keep only the evaluation of virulence by the pricking model.

As the reviewer pointed out, a previous study showed that a mutant unable to form biofilms in the fly's crop displayed a more virulent phenotype (Mulcahy, H *et al.* Plos Pathogens, 2011).). This is contrary to what should be expected considering the role of biofilms in *P. aeruginosa* pathogenicity. In the same study by Mulcahy *et al.* it is shown that the EPS elicits some components of the fly's innate immune response (antimicrobial peptides – AMPs), explaining why the EPS mutant is able to disseminate faster than the wild type strain. Probably the insect's immune system evolved to recognize the EPS as a pathogen associated molecular pattern (PAMP) and trigger the synthesis of AMPs to prevent oral infections with the ubiquitous *P. aeruginosa*, which cohabits many of the same ecological niches as insects. Therefore, we only see an apparent contradiction here because the EPS mutant is able to disseminate faster by avoiding the immune system does not preclude the importance of the biofilm for pathogenicity, even by the oral route. Thus, other biofilm deficient mutants, different from the EPS-lacking-strains, should be tested to validate this previous study. Specially, it would be better to test biofilm-deficient mutants that arise from the absence of factors other than those that can function as PAMP.

Thus, we consider the pricking model of infection more appropriate to evaluate the role of biofilm in the virulence of *P. aeruginosa*. This model more closely mimics natural infection by *P. aeruginosa*, which is not an enteropathogen. We want to emphasize that we consider the oral inoculation of *P. aeruginosa* a very useful model to directly visualize biofilms that develop in vivo over the epithelial tissues, but that the model has its limitations when evaluating the attenuation conferred by certain mutations, especially those affecting biofilm formation.

R1: The authors used one measure of biofilm formation to assess biofilms throughout, which is the presence of aggregates. It would strengthen the study to demonstrate that these aggregates had other features of biofilms, such as the presence of EPS and eDNA in the matrix, or increased antibiotic tolerance, or they somehow contributed to immune evasion.

A: We agree with the reviewer. Even though the methods we used are based on previously established models to assess biofilm, in the revised manuscript we now include new data showing that indeed the biofilm promoted by the oxylipins produces a matrix containing EPS and eDNA (now shown in page 4, lines 115-119 and Fig. 3B of the revised MS). A corresponding heading was added to the Methods section (page 11, lines 338-344).

R1: Since these reactions involving fatty acids require oxygen, would this pathway be operating in chronic lung infections in CF patients? CF lung infections are considered one of the best examples where biofilms contribute to a chronic infection.

A: While experimental evidence of the functionality of DS in CF patients chronically infected with *P. aeruginosa* is lacking, previous studies have detected the presence of abundant oxylipins in the sputum of these patients, revealing the feasibility of fatty acid oxygenation in that context (Jun Yang *et al.* Free Radic Biol Med, 2012). The wide distribution of oleic acid among human tissues also supports the feasibility of oxylipin production during *P. aeruginosa* infection in humans, including CF patients. We now have expanded the discussion of this topic in the revised manuscript (Discussion section, page 7-8, lines 215-238). We want to remark at this point that based on our data, the main role of oxylipins produced by *P. aeruginosa* could be during the initiation of the infectious process where it promotes the transition of the bacteria from a free swimming planktonic to a sessile hyper-twitching phenotype, as we propose in the model in figure 7. However, as the biofilm lifestyle during chronic infection implicates a cycle in which dispersion and formation of the de novo biofilm is constantly occurring, oxylipins also could play a role by regulating the bacterial colonization of new niches in the lung during the spread of the infection.

Answers to reviewer 2:

R2: They argue that the bacteria transform fatty acids scavenged from the host into oxygenated derivatives as a way to sense the host environment in order to induce appropriate bacterial responses. The finding that the two genes involved in oxylipin biosynthesis (PA2077 and 2078) are transcribed also under normal lab conditions (as can be seen in published transcriptional analysis) raises the question of whether *P. aeruginosa* is dependent on the external supply of fatty acids to produce oxylipins. Testing this would be important to validate the hypothesis.

A: This is an astute question on the part of the reviewer. In fact, this question has been addressed and published before (Martinez E. *et al.* JBC, 2011). *P. aeruginosa* is dependent on the external supply of fatty acids to induce the genes encoding the diol synthase enzymes necessary to produce oxylipins. In fact, the genes involved in diol synthase activity were initially identified by analysis of mutants of periplasmic enzymes induced by oleic acid (Estupiñan *et al.* BBA, 2014 *et al.*). The selection was based on a previous transcriptional study (Nouwens A.S. *et al.*,

Microbiology, 2003). In agreement, we have observed that a semi-purified protein extract of *P. aeruginosa* has diol synthase activity only if the bacterial culture is supplemented with oleic acid.

R2: This is a good manuscript that provides insight into the role of oxylipins in *P. aeruginosa* pathogenicity. However, there is no attempt to mechanistically unravel the link between oxylipin production and pathogenicity, and thus lacks broad appeal in its current state.

A: Our results, for the first time, provide strong evidence supporting an important physiological role for prokaryotic oxylipins in regulating bacterial physiology and pathogenicity, though admittedly the underlying mechanism has yet to be completely elucidated. However, we clearly demonstrated that part of the mechanism linking oxylipin production and pathogenicity is the negative regulation of flagellum-driven motility (swimming and swarming) and upregulation of twitching motility, which is known to promote attachment and biofilm formation. In turn, it is well documented that attachment of *P. aeruginosa* to a surface induces the expression of several virulence factors (Siryaporn A. et al, Proc Natl Acad Sci U S A, 2014 Nov 25;111(47):16860-5. doi: 10.1073/pnas.1415712111) and the role of biofilms in pathogenicity of *P. aeruginosa* is very well established and universally accepted. Although we have not unveiled the entire mechanism yet, we do, however, consider that our findings describe an entirely novel and unexplored regulatory signaling pathway governing bacterial-host interactions, which should be appealing to a broad readership in different fields of bacteriology. Furthermore, mechanistic studies are currently being conducted; however, their complexity and time-consuming nature preclude us from waiting for their conclusion and incorporation into the current study. Finally, we also do not believe the absence of a fully elucidated mechanism detracts from the message we want to convey on the importance of oxylipins in *P. aeruginosa* physiology.

Answers to reviewer 3:

R3: The authors mention that they determined the concentration of oxylipins produced during infection of *Drosophila*, however they make no mention of this physiological concentration and it is unclear if this concentration has any significance when studying the effect in vitro. Are the in vitro concentrations used in the same range as the in vivo concentrations and if not, why not?

A: Currently, we are in the process of standardizing the conditions to study the full lipidome of *Drosophila* in response to infection with the WT and Δ DS mutant. We plan to quantify the relative concentrations of all oxylipins detected in the flies. Thus, we inadvertently included the quantification protocol in the Methods section. However, for this paper we decided just to determine if the compounds are produced during the infection process, since we think that the average value of oxylipin concentration per fly lacks a physiological meaning since it would not take into consideration the spatiotemporal concentration of oxylipins produced during infection. First, considering that the diol synthase activity occurs in the bacterial periplasm and from there the derived oxylipins go to the extracellular medium it is highly probable that the concentration of oxylipins is locally higher at the site of the infection than in distant tissues, thus forming a microgradient. Additionally, it should also be considered that the concentration of oxylipins, like other signal molecules could drastically fluctuate during the course of the infection; so, concentrations will depend on the time point at which sample collection is done. In agreement with this notion we have observed that oxylipin concentrations rapidly decrease during the course of our experiments in vitro, suggesting that a great portion of the initial amount of oxylipins used in the experiments is transformed into other compounds or simply consumed by the β -oxidation pathway. Finally, we want to remark that the concentration of oxylipins we tested in vitro to evaluate microcolony formation are below the concentrations *P. aeruginosa* is able to

accumulate during in vitro culture (up to 10 mg/mL), depending on the availability of the substrate oleic acid and initial bacterium inoculum.

R3: The authors should mention in their materials and methods section how many times an experiment was repeated and how the significance values were determined.

A: All experiments were repeated at least three times. Now we have stated this information in each figure's legend and in the Statistical subsection of the Methods, as suggested by the reviewer.

R3: For the formation of microcolonies in vitro, the authors should include the wild type strain supplemented with oleic acid to compare the effect with the mutant. Here the physiological concentration of oxylipins may also be important when supplementing the mutant.

A: In order to detect the role of oxylipins on microcolony and early biofilm formation we performed experiments starting with a relatively small amount of bacteria ($OD \approx 0.05$) and the experiment was followed for a maximum of 3 hours. Using higher amounts of bacteria or longer incubation times the surface starts to become completely covered by attached bacteria making differences between conditions difficult to observe. As expected, under the conditions tested oxylipins were not detected in the media and consequently no difference was observed between PAO1 and the ΔDS mutant without addition of exogenous oxylipins. In supplemental data we originally provided a time-lapse experiment of oxylipin accumulation in the extracellular medium of PAO1 cultures. It should be noted that even when the initial quantity of bacteria in the suspension was set at $OD \approx 1$, oxylipins started to accumulate at approximately 3 hours (Fig. S8). However, we want to remark that during longer experiments the difference between PAO1 and the DS mutant in terms of biofilm formation is consistently significant, as shown in figure 2A of our manuscript. Below we are including a representative picture corresponding to the experiments that provided the data for Fig. 2A, where the reviewer can clearly appreciate the difference in biofilm formation between PAO1 and DS in the presence of oleic acid after 16 h of incubation.

R3: Similarly, for the in vivo experiments, the wild type can be included in order to compare the extent of microcolony formation.

A: We have now included these results in figures 5 and S6, and modified the text of the corresponding heading of the Results section in the revised manuscript (page 5, lines 143-154). We would like to remark here that as explained above for reviewer 1, contrary to what is observed in the pricking model of inoculation, no oxylipins seem to be produced by *P. aeruginosa* in *D. melanogaster* crops and consequently no differences were observed between PAO1 and DS in terms of biofilm formation.

R3: No results for bacterial attachment and kinetics (0 to 3 h) are presented.

A: The reviewer noticed we mentioned in the Methods section that we performed a time-course experiment for microcolony formation from 0 to 3h. Indeed we did this experiment and we now provide the results below in which the reviewer can see a time course of PAO1 attachment in the presence of 10-HOME. However, we excluded this data from the manuscript because we considered it irrelevant to the story and we inadvertently forgot to remove the accompanying information from the Methods section. We still feel that the kinetic analysis contributes little and does not alter the overall conclusions drawn from our findings. Thus, we removed this statement from the Methods section.

R3: Minor issue: For the oral inoculation of *Drosophila* – only the GFP-expressing strain’s result is given, so the authors may consider omitting the mCherry from the materials and methods (Ln 316)

The authors should also look at some minor language issues:

Ln 71: Bacterial motility.....

Ln 75: “...motility has consequences...”

Ln 117: “...promoted microcolony formation...”

Ln 201: “...transform fatty acids scavenged...”

Ln 223: “For this plasmid...”

Ln 228-229: “...by replacing the mutated allele with the original...”

Ln 238: *Escherichia coli* has to be in italics

Ln 288: “...software Cell Sense...”

Ln 311: I do not think “suffered” is the correct word in this context

Ln 338: “...determine the concentrations of the analytes...”

Ln 357: *Drosophila* must be in italics

Reference 16 – *Pseudomonas* should be with a capital letter.

A: All minor issues detected by the reviewer were corrected.

REVIEWERS' COMMENTS:

Reviewer #1 (Remarks to the Author):

The authors have considered the initial reviewer comments and made a satisfactory response. However, I do have a few comments.

The authors now show that the biofilm phenotypes of both the wt and ds mutant are enhanced in the crop with exogenous oxylipins. They mention in the rebuttal but not in the paper, that there were no differences in virulence in the feeding model. It is worth adding that the virulence and biofilm experiments were both performed in the feeding model.

The authors make an argument in the rebuttal that the pricking model more closely resembles a natural *Drosophila* infection by *Pseudomonas aeruginosa*, which is not an enteropathogen. This is not a convincing argument, how does *Pseudomonas* get introduced into the fly cavity in a normal encounter? While *Pa* is not a rapid killing enteropathogen, I would argue it still is a slow killing enteropathogen of the fly. My main point is that you shouldn't argue that the model where you have interesting data is the best model, but rather to state the pricking model was more useful to demonstrate the role of oxylipins and biofilm formation.

It is now apparent to me that in this study, biofilms are not formed in the crop within 20 hrs, while the original PLoS Pathogens article (Mulcahy et al) did report biofilm aggregates within the crop at 24 hrs. While this study shows that biofilms are robust after feeding the flies oxylipins, the discrepancy in normal feeding infections should be commented on. The bacteria were grown in almost identical conditions before and during fly feeding, however, the flies were reared on different media. It may be that oxylipins are available in the crop at early time points when flies are reared on different growth media.

Reviewer #3 (Remarks to the Author):

I recommend that the revised manuscript be published.

Point-by-point discussion of issues raised by reviewer #1:

R1: The authors have considered the initial reviewer comments and made a satisfactory response. However, I do have a few comments.

The authors now show that the biofilm phenotypes of both the wt and ds mutant are enhanced in the crop with exogenous oxylipins. They mention in the rebuttal but not in the paper, that there were no differences in virulence in the feeding model. It is worth adding that the virulence and biofilm experiments were both performed in the feeding model.

A: The virulence experiment through the oral route has now been included in the revised manuscript as suggested by the reviewer (pag. 7 first and second paragraphs of section “Oxylipins promote *P. aeruginosa* virulence in *Drosophila*”).

R1: The authors make an argument in the rebuttal that the pricking model more closely resembles a natural *Drosophila* infection by *Pseudomonas aeruginosa*, which is not an enteropathogen. This is not a convincing argument, how does *Pseudomonas* get introduced into the fly cavity in a normal encounter? While Pa is not a rapid killing enteropathogen, I would argue it still is a slow killing enteropathogen of the fly. My main point is that you shouldn't argue that the model where you have interesting data is the best model, but rather to state the pricking model was more useful to demonstrate the role of oxylipins and biofilm formation.

A: We wanted to express that *pseudomonas* is an opportunistic pathogen that takes advantage of immunocompromised or somehow weakened individuals (wounded, burned, etc.) to produce an infection. For example, the majority of human cases of gastrointestinal (GI) infections caused by PA occur in cancer patients subjected to chemotherapy. In these cases PA can infect anywhere, not only the GI tract. We agree that “natural” was not the proper word to describe the *Drosophila* infection since natural is anything occurring in nature disregarding how common or not an event is (even immunocompromised/wounded individuals are very natural). We should say instead “normal” or “common”. *Pseudomonas aeruginosa* is not classified as an enteropathogen in the classical sense of this term (at least we haven't found any report considering PA as an enteropathogen). PA could be a slow killing enteropathogen in lab conditions, but in nature these conditions are very different. In nature, *D. melanogaster* is primarily associated with rotting fruits, but also with a wide variety of decaying vegetables and other plant matter (DOI:

10.7554/eLife.06793.001, *The Secret life of Drosophila Flies*). *Drosophila*'s decaying host food is also home to many microbes, including PA, which is known to be a common colonizer of many fruits and green plants (*Plant Physiol.* 2000 Dec; 124(4): 1766–1774). We also think that PA can easily get introduced into internal tissues of wounded insects, which are indeed very common in nature.

Furthermore, in our previous response we didn't state that the pricking model was the "best". What we textually stated was: "*Since there is no difference in the capacity of PAO1 and Δ DS to form a biofilm in the fly crops, neither oxylipins are produced in the digestive tract, we drew the conclusion that the feeding model is not useful to evaluate differences in virulence between PAO1 and Δ DS. As we are interested in precisely knowing the role of oxylipins in virulence we decided to exclude this result from the paper and keep only the evaluation of virulence by the pricking model*". Later on we stated: "*Thus, we consider the pricking model of infection more appropriate to evaluate the role of biofilm in the virulence of *P. aeruginosa**". This doesn't mean that the oral model is inferior, but just that it was not useful for our specific purposes. In fact we also said: "*We want to emphasize that we consider the oral inoculation of *P. aeruginosa* a very useful model to directly visualize biofilms that develop in vivo over the epithelial tissues, but that the model has its limitations when evaluating the attenuation conferred by certain mutations, especially those affecting biofilm formation*". The oral model can still be useful to test other factors effect on virulence. However, we found that most reports studying PA virulence in *Drosophila* used the pricking model, even among those citing the Mulcahy et al paper.

R1: It is now apparent to me that in this study, biofilms are not formed in the crop within 20 hrs, while the original PLoS Pathogens article (Mulcahy et al) did report biofilm aggregates within the crop at 24 hrs. While this study shows that biofilms are robust after feeding the flies oxylipins, the discrepancy in normal feeding infections should be commented on. The bacteria were grown in almost identical conditions before and during fly feeding, however, the flies were reared on different media. It may be that oxylipins are available in the crop at early time points when flies are reared on different growth media.

A: We want to remark here that we were, in fact, able to observe bacterial aggregates at 20 hrs similar to those observed by Mulcahy et al in their report at 24 hrs (see below Fig. 1 A-D). However, after the addition of oxylipins, the difference in results is dramatic (Fig. 1 E and F). We preferred to use PA expressing a GFP because we noticed that bacterial cells looked better defined than using mCherry-expressing-PA and also because red fluorescence bleached faster than the green one. This may explain why we observed more free bacteria dispersed throughout the crop than it is seen in Mulcahy's picture (see below B). Or could be simply that at 24 hrs bacteria are more grouped into microcolonies than at 20 hrs or that different brightness/contrast setting were used to take the pictures.

Figure 1. Crop colonization of *D. melanogaster* using our study conditions compared to those used by Mulcahy et al. A) Picture of a crop (different from the one we showed in Fig. 6A of our MS) compared to the picture by Mulcahy et al (panel B), which show colonization by PAO1 in the absence of oxylipins at 20 and 24 h, respectively. In C) and D) we show the same panels of Fig. 6A and supplemental Fig. 5A and in E) and F) crops images in the presence of 10-HOME for an easier appreciation in comparison with Mulcahy's results.

Reviewer #3 (Remarks to the Author):

R1: I recommend that the revised manuscript be published.

A: We appreciate Reviewer #3 considers our paper adequate for publication.